# Widespread chromatin context-dependencies of DNA double-strand break repair proteins

Xabier Vergara [1,2,3,4], Anna G. Manjón[3,4,10], Marcel de Haas[1,2,4,10], Ben Morris[5], Ruben Schep[1,4], Christ Leemans[1,4], Anoek Friskes [3,4], Roderick L. Beijersbergen [5,6], Mathijs A. Sanders[7,8], René H. Medema [3,4] ✉ & Bas van Steensel [1,2,4,9] ✉

DNA double-strand breaks are repaired by multiple pathways, including non-homologous end-joining (NHEJ) and microhomology-mediated end-joining (MMEJ). The balance of these pathways is dependent on the local chromatin context, but the underlying mechanisms are poorly understood. By combining knockout screening with a dual MMEJ:NHEJ reporter inserted in 19 different chromatin environments, we identified dozens of DNA repair proteins that modulate pathway balance dependent on the local chromatin state. Proteins that favor NHEJ mostly synergize with euchromatin, while proteins that favor MMEJ generally synergize with distinct types of heterochromatin. Examples of the former are BRCA2 and POLL, and of the latter the FANC complex and ATM. Moreover, in a diversity of human cancer types, loss of several of these proteins alters the distribution of pathway-specific mutations between heterochromatin and euchromatin. Together, these results uncover a complex network of proteins that regulate MMEJ:NHEJ balance in a chromatin context-dependent manner.

DNA double-strand breaks (DSB) are repaired by multiple repair pathways such as non-homologous end-joining (NHEJ), homologous recombination (HR) and microhomology-mediated end joining (MMEJ). These pathways act in an equilibrium that is referred to as the DNA repair pathway balance (reviewed in ref. [1]). Defects in this balance can compromise genome stability, but also offer opportunities for therapy, particularly in cancer[2]. Pathway balance is influenced by several factors, including cell cycle[3], break complexity[4] and the chromatin context in which a DSB occurs[5,6]. The latter is generally attributed to molecular interactions between specific repair proteins and distinct chromatin proteins, in some instances regulated by posttranslational modifications[7–9]. Such local interactions can alter the recruitment of the repair protein to a DSB, or modulate its activity in the repair process. Yet, the overall extent and the principles of this interplay between chromatin and repair proteins have remained poorly studied.

Here, by screening hundreds of DNA repair proteins, we uncover that chromatin context has a widespread influence on the relative contribution of specific DNA repair proteins to repair pathway balance.

[1]Division of Gene Regulation, Netherlands Cancer Institute, Amsterdam, The Netherlands. [2]Division of Molecular Genetics, Netherlands Cancer Institute, Amsterdam, The Netherlands. [3]Division of Cell Biology, Netherlands Cancer Institute, Amsterdam, The Netherlands. [4]Oncode Institute, Utrecht, The Netherlands. [5]NKI Robotics and Screening Center, Netherlands Cancer Institute, Amsterdam, The Netherlands. [6]Division of Molecular Carcinogenesis, Netherlands Cancer Institute, Amsterdam, The Netherlands. [7]Department of Hematology, Erasmus MC Cancer Institute, Rotterdam, The Netherlands. [8]Cancer, Ageing and Somatic Mutation (CASM), Wellcome Sanger Institute, Hinxton, UK. [9]Department of Cell Biology, Erasmus University Medical Center, Rotterdam, The Netherlands. [10]These authors contributed equally: Anna G. Manjón, Marcel de Haas. ✉e-mail: r.medema@nki.nl; b.v.steensel@nki.nl

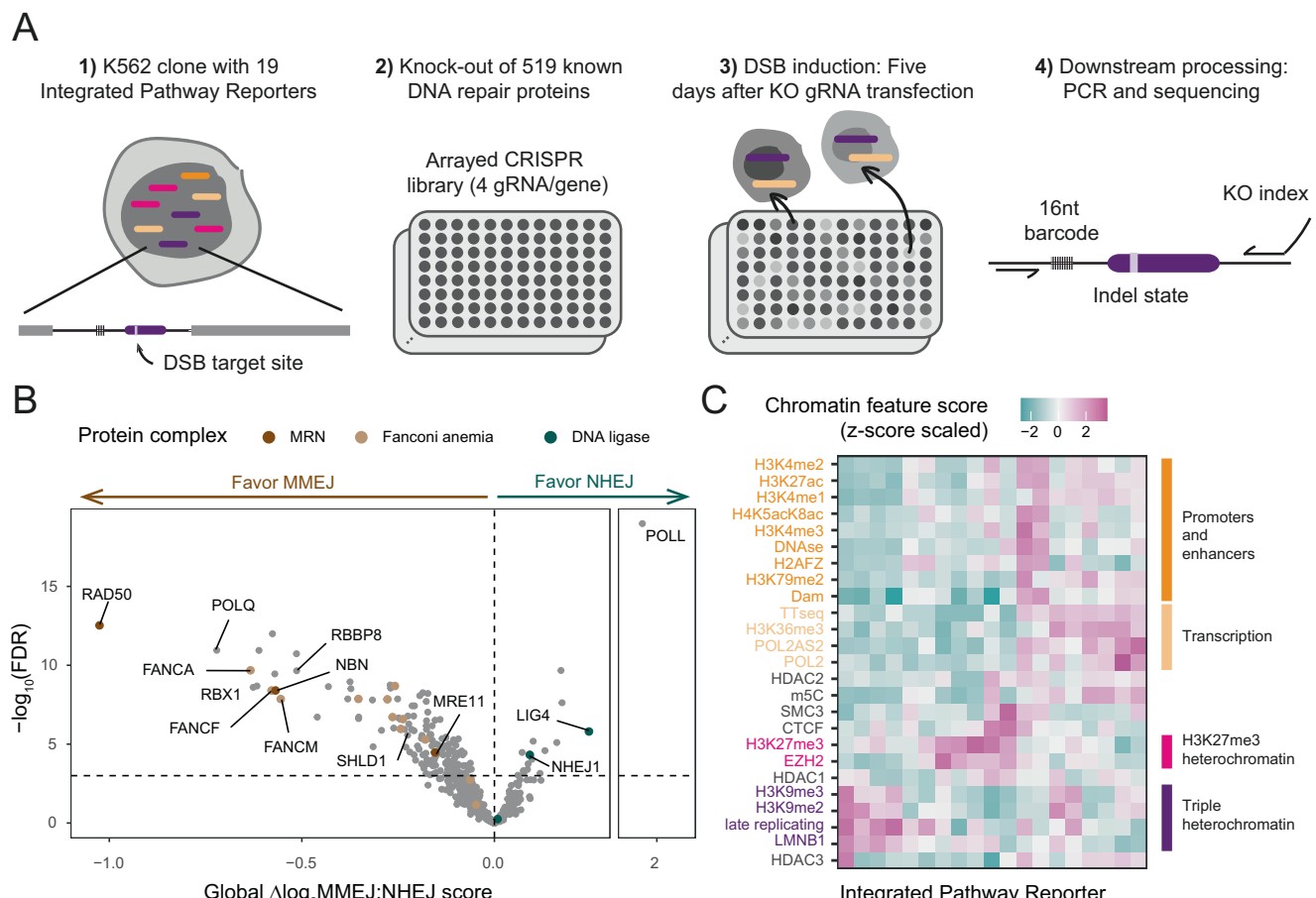

**Fig. 1 | Multiplexed CRISPR screen to assess chromatin context-dependencies of DNA repair proteins. A** Overview of screen design. See main text for explanation. **B** Volcano-plot of global $\Delta\log_2$MMEJ:NHEJ scores (mean value of 19 IPRs). Horizontal dotted line shows significance threshold (FDR = 0.001). Labels mark proteins highlighted in the text. **C** Heatmap of levels of 25 chromatin features at the 19 IPR sites. Four major chromatin states and their defining features are highlighted in distinct colors. Source data are provided as a Source Data file.

## Results

### Experimental design

We focused on the balance between NHEJ and MMEJ, which are two of the main mutagenic DSB repair pathways, particularly for DSBs generated during CRISPR editing[10]. We applied a sequencing-based assay that determines the MMEJ:NHEJ balance after induction of a DSB by Cas9, with high accuracy and in multiple genomic loci in parallel[6]. For this we employed a human K562 cell line with 19 barcoded Integrated Pathway Reporters (IPRs) inserted throughout the genome (Fig. 1A). Importantly, the integration sites represent all major known chromatin types[6] (see below). In this cell line we conducted three biological replicates of a 96-well CRISPR/Cas9 screen to knock out (KO) 519 proteins that had previously been linked to at least one DNA repair pathway (Fig. 1A and Supplementary Fig. 1A; Supplementary Data 1 and Supplementary Table 1; Detailed protocol in Supplementary Methods). For each KO we then induced a DSB in all IPRs; after 72 h to allow repair to occur, we isolated genomic DNA and sequenced the IPRs to determine the MMEJ:NHEJ balance as the ratio between the signature indels +1 (NHEJ$_{ins}$) and −7 (MMEJ$_{del}$) (Supplementary Table 2)[6]. For each IPR–KO combination we then computed the $\log_2$ fold change in MMEJ:NHEJ balance [$\Delta\log_2$MMEJ:NHEJ] relative to the average of a set of 33 mock KO control samples (in which gRNA was omitted in the KO step). We averaged the results of three replicates, resulting in a 519 × 19 matrix of $\Delta\log_2$MMEJ:NHEJ scores (Supplementary Data 2). These scores reflect the contribution of each tested protein to the MMEJ:NHEJ balance in 19 well-characterized chromatin contexts (Supplementary Fig. 1B).

### Repair proteins affecting pathway balance globally

We first assessed the impact of the tested proteins on the global MMEJ:NHEJ balance, i.e., irrespective of the local chromatin context, by evaluating the mean $\Delta\log_2$MMEJ:NHEJ scores of the 19 IPRs. At an estimated false discovery rate (FDR) of 0.001, 149 proteins *favored MMEJ* (Fig. 1B; Supplementary Data 5), i.e., these proteins either are required for full MMEJ activity or they inhibit NHEJ when present. Among these are known key components of the MMEJ pathway, such as POLθ (POLQ), proteins of the MRN complex and CtIP (RBBP8). We also found that several Fanconi anemia (FA) proteins (e.g., FANCA, FANCF, FANCM, FANCD2), which are central proteins of inter-strand crosslink (ICL) repair, favored MMEJ. Unexpectedly, proteins that either directly (SHLD1[11]) or indirectly (RBX1[12]) limit long-range resection, a key step for HR, favored MMEJ. This suggests that limitation of long-range resection favors MMEJ over other pathways. Conversely, 16 proteins favored NHEJ globally, including known components of the NHEJ pathway, such as Ligase IV (LIG4)[13,14], XLF (NHEJ1)[15,16] and DNA polymerase lambda (POLL)[17]. Thus, the screen confirmed several known key proteins in the repair of DSBs generated by Cas9 and other nucleases[17,18].

### Many repair proteins show significant chromatin context-dependency

Next, we asked which proteins exhibited chromatin context-dependency (CCD) of their $\Delta\log_2$MMEJ:NHEJ scores across the 19 IPRs. As we and others previously demonstrated, integrated

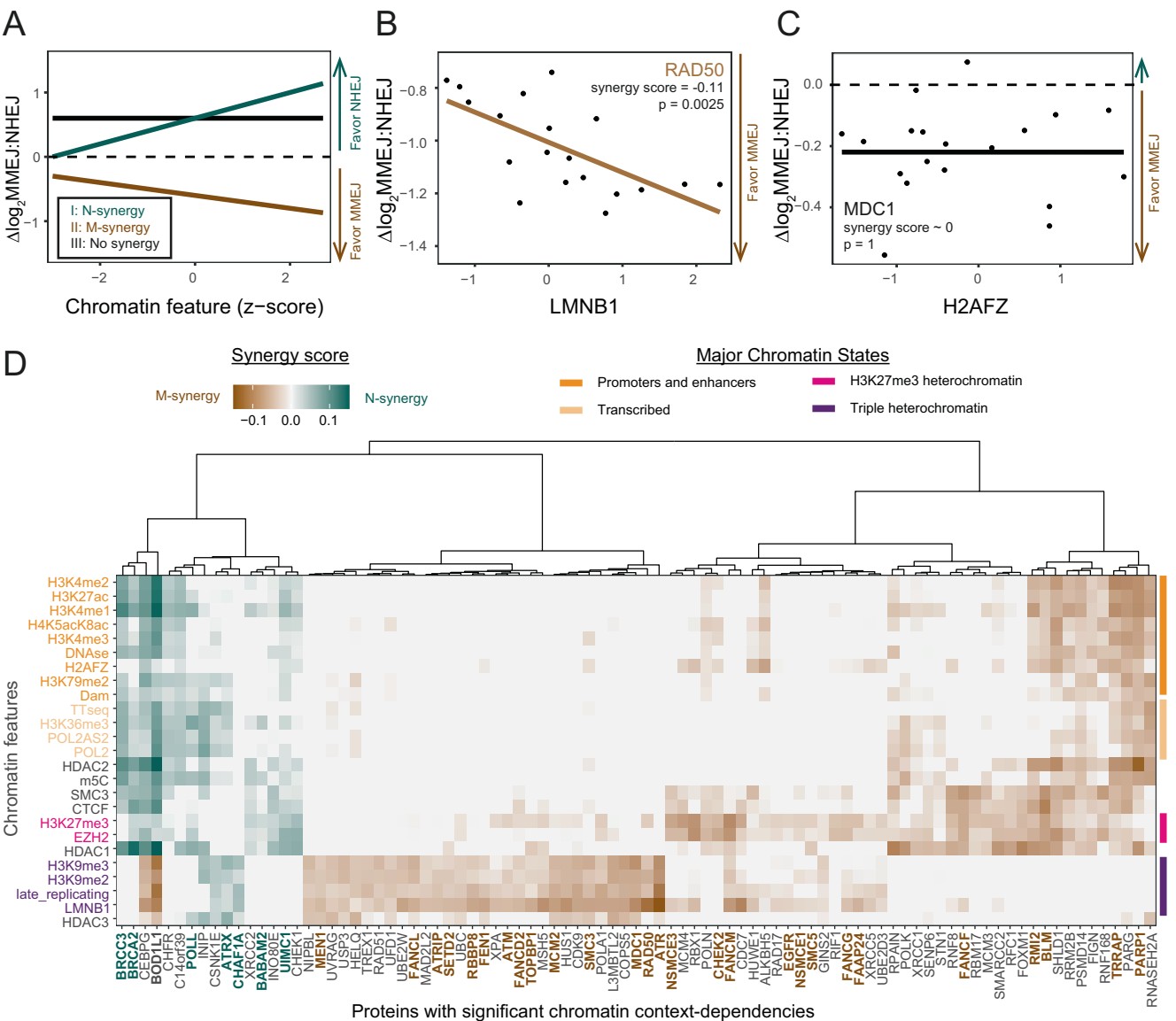

**Fig. 2 | CCDs of 89 DNA repair proteins. A** Illustration of M- and N-synergy concepts. For chromatin feature−protein combinations with N-synergy (I in green) $\Delta\log_2$MMEJ:NHEJ scores increase as the chromatin feature levels increase. For combinations with M-synergy (II in brown) $\Delta\log_2$MMEJ:NHEJ scores decrease as the chromatin feature levels increase. For combinations with no synergy (III in black) $\Delta\log_2$MMEJ:NHEJ scores do not correlate with the individual chromatin features. **B** M-synergy example: linear fit (brown line) of RAD50 $\Delta\log_2$MMEJ:NHEJ scores with LMNB1 interaction levels. Slope (synergy score) and *p*-value of the correlation are shown in the graph. **C** No synergy example: linear fit (black line) of MDC1 $\Delta\log_2$MMEJ:NHEJ scores with H2AFZ levels. Slope (synergy score) and two-sided *p*-value of the correlation are shown in the graph. **D** Heatmap of synergy scores of all 89 proteins with significant CCDs. Proteins (columns) mentioned in the text are highlighted in bold. Chromatin features (rows) are colored and ordered as in Fig. 1C. Source data are provided as a Source Data file.

reporters generally adopt the local chromatin state[6,19–21]. We therefore used a set of high-quality epigenome maps from K562 cells (Supplementary Data 3) to infer the levels of 25 chromatin features on each of the IPRs (Fig. 1C, Supplementary Data 4). We then applied a three-step linear modeling approach (see Supplementary Methods, Supplementary Figs. 2 and 3) to identify proteins for which the $\Delta\log_2$MMEJ:NHEJ scores correlated significantly with one or more chromatin features. According to this analysis, 89 (17.1%) of all tested proteins showed a significant CCD at 5% FDR cutoff. Of 33 mock KO samples only one (3%) passed this cutoff, confirming the low rate of false positives. These results indicate that a surprisingly large proportion of DNA repair proteins modulate the MMEJ:NHEJ balance with a significant CCD (Supplementary Data 5).

**Distinct patterns of synergies**

Next, for each of the identified proteins we asked which chromatin features explain the CCD. For this we considered the slope of linear fits that correlate $\Delta\log_2$MMEJ:NHEJ scores with each individual chromatin feature (see Supplementary Methods, Supplementary Fig. 4). A synergy score (slope) is positive when the repair protein favors NHEJ with increasing levels of the chromatin feature (I in Fig. 2A). We will refer to this as "N-synergy". When the synergy score is negative, the protein favors MMEJ with increasing levels of the chromatin feature (II in Fig. 2A); this we will refer to as "M-synergy". For example, we found that RAD50 is M-synergistic with Lamin B1 (LMNB1) (Fig. 2B and Supplementary Fig. 4D), indicating that RAD50 favors MMEJ preferentially in regions that interact with the nuclear lamina. A synergy score near zero points to a lack of detectable synergy of the tested pair (III in

Fig. 2A), as exemplified by the repair protein MDC1 and the chromatin feature H2AFZ (Fig. 2C).

## M- and N-synergies: distinct distributions across chromatin types

Hierarchical clustering of the synergy scores of all 89 proteins with significant CCDs revealed striking patterns (Fig. 2D). First, 16 proteins have N-synergies while 75 have M-synergies, with two proteins overlapping due to mixed synergies. Thus, proteins with M-synergies are much more prevalent than proteins with N-synergies. 11 out of 12 proteins with CCDs that are annotated as MMEJ and/or NHEJ proteins (GO:0006303, Supplementary Fig. 5A), exhibit M-synergy and only 1 N-synergy. This may reflect a higher complexity of the MMEJ pathway compared to the NHEJ pathway[17,18,22]. Second, N-synergies predominantly involve euchromatic features, such as marks of active promoters and enhancers (e.g., H3K4me3 and H3K27ac) and transcription activity (e.g., TT-seq, POL2 and H3K36me3) (Fig. 2D). Only a few proteins show N-synergy with heterochromatin, either alone or in combination with a subset of euchromatic marks. Third, M-synergies are divided over three main clusters, with prominent roles for distinct classes of heterochromatin. One cluster of 33 proteins has consistent M-synergy with heterochromatin that is marked by a combination of H3K9me2/3, late replication and interactions with LMNB1. We will refer to this type of heterochromatin as triple heterochromatin. We find in this cluster proteins involved in DSB processing (GO:000729), single-strand annealing (GO:0045002) and meiotic recombination (GO:1990918) (Supplementary Fig. 5B–D). A second cluster of 31 proteins is primarily M-synergistic with H3K27me3-marked heterochromatin, often combined with LMNB1; and a third cluster of 11 proteins shows M-synergy with various euchromatin marks, frequently combined with H3K27me3. Thus, the vast majority of M-synergies involve either triple or H3K27me3 heterochromatin, unlike most N-synergies. (Fig. 2D). The skewed distribution of M- and N-synergies between heterochromatin and euchromatin provides an explanation for the earlier observation that the MMEJ:NHEJ ratio tends to be higher in heterochromatin[6]. Interestingly, we also identify proteins that favor MMEJ in euchromatin (PARP1, BLM and RMI2). Such proteins might be key to ensure that MMEJ carried out throughout the genome rather than being restricted to heterochromatin.

## CCD effects compared to global effects

Of the 89 proteins with significant CCDs in K562, 46 modulate MMEJ:NHEJ balance globally with preferential impact on specific chromatin contexts (e.g. RAD50, FANCM or ATM). In these cases, CCD and global effects tend to have similar effect sizes (Supplementary Fig. 6, see Supplementary Methods). Additionally, 43 proteins only modulate MMEJ:NHEJ balance in specific chromatin contexts (Supplementary Fig. 6E). Thus, the magnitude of CCD effects is often similar or larger than the chromatin-independent contributions of individual proteins.

## Incomplete penetrance of the screen

In our screen, we estimated that individual alleles of each targeted gene were disrupted with an efficiency of roughly 60% (Fig. S7A, B). Thus, KO of individual proteins was incomplete, and hence negative results should be interpreted with caution. For instance, BRCA1 did not show an effect on the global MMEJ:NHEJ ratio (Supplementary Data 5), unlike what we reported previously in the same cell line[6] (Fig. S7C). Lack of CCD of BRCA1 could thus have been a false-negative result. Indeed, re-analysis of the published data indicates that BRCA1 in fact exhibits modest but detectable M-synergy with heterochromatin (Fig. S7D). In the same previous study, RAD51, BRCA2 and POLQ were depleted by RNA interference with an efficiency of ~75%[6]. Analysis of these earlier data reveals CCD patterns that are highly similar to our screen results (Fig. S7D), but both the global effects and the synergy

scores are on average 2.2-fold higher (Fig. S7E, F). Several other lines of evidence (Supplementary Fig. 7G–L) also indicate that the CCD effect sizes in the screen are substantially underestimated due to incomplete KO of each protein.

## Integrated pathway reporters in RPE-1 cells

To test whether these CCDs also occur in other cell types, we repeated these analyses for a subset of proteins in retinal pigment epithelium (RPE-1) cells. We initially chose three available variants of this cell line[23,24]: wild-type (WT), knockout of p53 (p53[KO]) and double knockout of p53 and BRCA1 (p53/BRCA1[dKO]). BRCA1 normally directs part of DSB repair activity towards HR[25] and shows a weak M-synergy with heterochromatin in K562 cells (Supplementary Fig. 7D). Other HR proteins (GO:0000724), such as ATM, RBBP8 and BRCA2, exhibit M- or N-synergies with different chromatin contexts (Supplementary Fig. 5E), and hence we investigated whether their CCD pattern would be affected by the absence of BRCA1. From each of these lines we established polyclonal cell pools carrying randomly integrated IPRs; we mapped the integration sites, and determined the levels of several key chromatin features at these integration sites. In these cell pools we then knocked out 20 proteins that exhibited CCDs in K562 cells (Supplementary Fig. 8, see Methods). Next, we activated Cas9 and measured the relative activities of MMEJ and NHEJ in each IPR. Unfortunately, the WT cell pool exhibited strong clonal drift that caused rapid loss of most IPRs, compromising statistical power of the CCD analysis; moreover, transfections of the WT cells triggered morphological changes that pointed to a strong stress response. Below we therefore only present results from the two RPE-1 lines lacking p53.

## CCDs in RPE-1 cells

We first assessed global $\Delta\log_2$MMEJ:NHEJ scores in p53[KO] and p53/BRCA1[dKO] cells. Overall, these global effects are very similar to those measured in K562 (Fig. 3A, B). We then assessed CCDs in the two RPE-1 cell lines. Of the 20 tested proteins, 16 exhibited M-synergy in at least one of the two RPE-1 lines (at FDR < 0.25). Of these, 15 also showed M-synergy in K562 cells (Fig. 3C). The only protein in RPE-1 cells with N-synergy is POLL, which shows a N-synergy in p53/BRCA1[dKO] cells (Fig. 3C). Two other proteins with N-synergies in K562, CHAF1A or BOD1L1, exhibit M-synergies in RPE-1 cells. Interestingly, these two proteins are chromatin proteins and might regulate MMEJ:NHEJ balance indirectly through changes in chromatin. Next, we checked the similarity of CCD *patterns* between the cell types. In both RPE-1 cell lines, CCD patterns are more similar to K562 than expected by random chance (Fig. 3D).

Compared to K562, PARP1, BLM and RMI2, which exhibit M-synergies with euchromatic features, show almost identical CCD patterns in p53[KO] (Fig. 3E), while this similarity is reduced in p53/BRCA1[dKO] cells. (Fig. 3F). These differences may reflect cell-type specific regulatory mechanisms, but we cannot rule out that the different IPR integration sites (sampling partially different chromatin states) may account for some of the differences. Nevertheless, the results in RPE-1 cells confirm that CCDs are widespread and show overall similarities—but also some differences—between cell types.

## Interpretation of M- and N- synergies

We note that M-synergy does not necessarily imply that the protein locally boosts MMEJ; it may also locally suppress repair via an alternate repair pathway and thereby shift the balance. Similarly, N-synergy may be either due to local activation of NHEJ or local suppression of other repair pathways. Furthermore, we emphasize that the synergies as defined here do not necessarily imply a direct molecular link between the repair protein and the chromatin feature; the feature may also be a proxy for an unknown chromatin feature that is closely linked. For this reason, most of our analyses below focus on the major known chromatin states that are represented by one or more features in our dataset. We

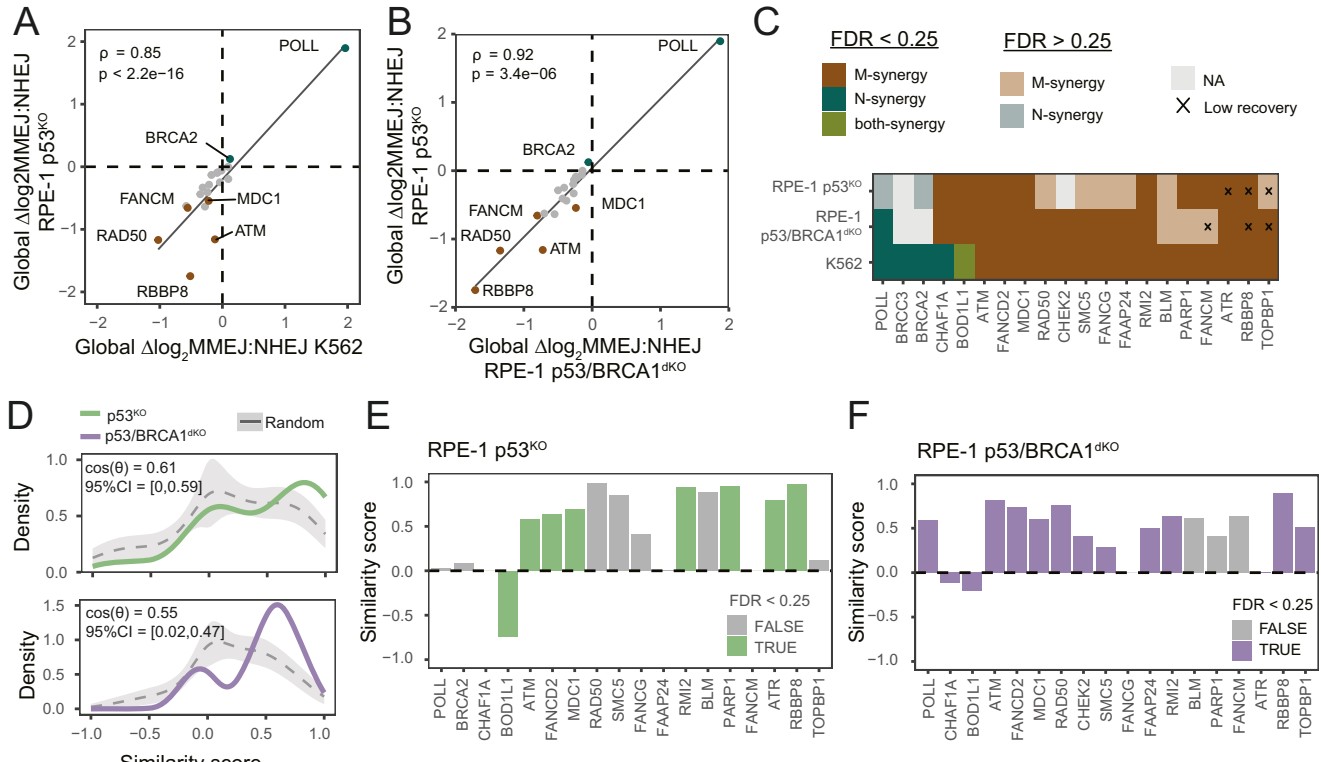

**Fig. 3 | CCDs of DNA repair proteins in RPE-1 cells. A** Scatter plot of global $\Delta\log_2$MMEJ:NHEJ scores in RPE-1 p53$^{KO}$ (y-axis) and K562 (x-axis). Spearman's rho and two-sided p-value of the correlation are shown in the graph. Proteins highlighted in the text are shown in brown (M-synergy) or in green (N-synergy). **B** Same as in A but for RPE-1 p53$^{KO}$ (y-axis) and RPE-1 p53/BRCA1$^{dKO}$ (y-axis). **C** M- and N-synergies in RPE-1 p53$^{KO}$, p53/BRCA1$^{dKO}$ and K562 cells of the tested 20 DNA repair proteins. M-synergies are shown in brown, N-synergies in green and both synergies in olive. Synergies detected with an FDR < 0.25 are shown in solid colors and synergies with FDR > 0.25 are shown in lighter colors. Empty values in gray represent DNA repair proteins that did not change the log$_2$MMEJ:NHEJ balance in RPE-1 cells. M- or N-synergies marked by a X were detected in samples with low DNA recovery compared to controls samples (See Methods and Supplementary Fig. 8A). **D** Distribution of pairwise similarity scores for CCDs patterns across chromatin features in K562 and RPE1 p53$^{KO}$ cells (top) and RPE1 p53/BRCA1$^{dKO}$ cells (bottom). In both plots, the green/purple distributions represent similarity scores of the same protein and in gray the distribution of random protein pairs (mean ± s.d. of 1000 draws). In the top left corner of each graph, the mean similarity score (cos(θ)) and 95%CI of the mean similarity scores of 1,000 random draws. **E, F** Pairwise similarity scores for CCD patterns of each of the proteins assayed in (**E**) RPE-1 p53$^{KO}$ cells (n = 18) and (**F**) RPE-1 p53/BRCA1$^{dKO}$ cells (n = 18). Columns in green or purple represent M- or N-synergies with an FDR < 0.25 and in gray synergies with an FDR > 0.25. Source data are provided as a Source Data file.

also note that some hits in our screen can be explained by indirect effects. For example, FOXM1 and EGFR are known to be regulators of various genes that encode DNA repair proteins[26,27], while there is no evidence that they directly mediate DNA repair. Below we highlight findings that are more likely to involve close interactions with chromatin.

### M-synergies of canonical MMEJ proteins
Among canonical components of the MMEJ pathway, several exhibit M-synergy in K562 and RPE-1 cells. This includes RAD50 (Fig. 2B), CtIP/RBBP8 and FEN1, which show exclusive M-synergy with triple heterochromatin; and PARP1 which has selective M-synergy with euchromatin and H3K27me3 (Fig. 2D).

### Proteins that interact tend to have similar CCD patterns
Some proteins that are part of the same complex, such as BLM and RMI2[28], show highly similar M-synergy (Fig. 2D). We asked whether this is a general trend among pairs of proteins that are known to physically interact in vivo according to the BioGRID database[29]. We identified n = 118 interacting pairs among proteins with significant CCDs (Supplementary Data 6). Similarities in CCD patterns were significantly higher between physically interacting proteins than expected by random sampling (Fig. 4A, B, empirical test p < 0.001). Among the 118 interacting pairs, we even found three 'cliques' of at least four proteins that are connected by pairwise physical interactions (Fig. 4C). One of these cliques encompasses ATM and its phosphorylation targets MDC1, TOPBP1,

and FANCD2. All these proteins show highly similar M-synergies with triple heterochromatin in K562 (Fig. 4D). This clique also shows similar M-synergies in p53$^{KO}$, p53/BRCA1$^{dKO}$ RPE-1 cells (Fig. 4G). TOPBP1-interacting proteins ATRIP and ATR also show M-synergies with triple heterochromatin. In line with this, ATM, TOPBP1, ATR and ATRIP have been previously linked to repair of heterochromatin DSBs[30,31].

### Heterochromatin M-synergy of the FANC complex
Additionally, we found a clique that consists of Fanconi anemia (FA) proteins (FANCF, FANCM, FANCG and FANCD2) and a third clique with two FA proteins together with BLM and RMI2 (Fig. 4E, F). These two cliques also show similar M-synergies in RPE-1 cells (Fig. 4H, I). Although FA proteins are primarily known to be involved in repair of inter-strand cross-links[32], they have also been implicated in MMEJ[18]. Six out of 12 tested FA proteins show selective M-synergies with either H3K27me3 or triple heterochromatin, or both. Moreover, four additional FA proteins (FANCA, FANCB, FANCC and FANCI) showed similar trends although they individually did not pass the significance threshold (Fig. 4K). These results indicate that the FA complex is an important regulator of MMEJ:NHEJ balance in heterochromatin.

### M-synergy of the SMC5/6 complex
Another complex implicated in DSB repair in heterochromatin is the SMC5/6 complex[31,33]. SMC5, NSE1 (NSMCE1) and NSE3 (NSMCE3) exhibit M-synergies with H3K27me3 and LMNB1. SMC6 displays similar

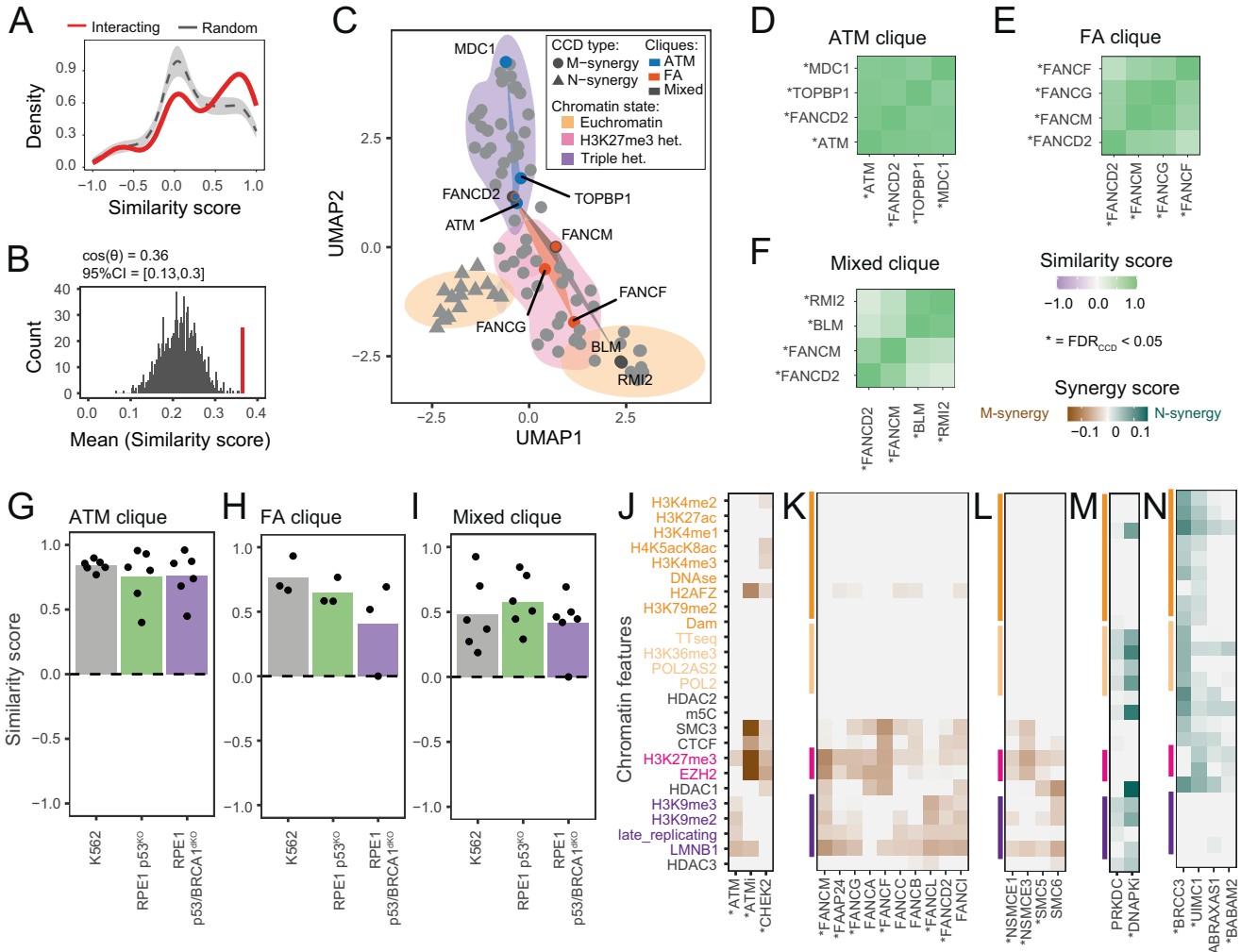

**Fig. 4 | Proteins that physically interact tend to have similar CCD patterns.**
**A** Distributions of pairwise similarity scores for CCD patterns across the 25 chromatin features, between interacting proteins (red; 118 pairs) and between randomly picked protein pairs (mean ± s.d. of 1,000 draws of 118 random pairs). **B** Mean similarity score of 118 interacting protein pairs (red line) compared to the distribution of mean similarity scores of 1000 random draws as in (**A**) (gray histogram), indicating that high similarities of CCD patterns of interacting protein pairs cannot be explained by random chance (two-sided $p$-value = 0.0006). **C** Uniform Manifold Approximation and Projection (UMAP) visualization of proteins with CCDs. Each dot represents a protein, with the shape indicating the type of synergy. Color clouds show the major chromatin state that explains each CCD. Three

'cliques' of four interacting proteins are shown as colored quadrangles. Proteins shared between multiple cliques are marked by concentric circles with the color of each clique they are part of (**D**–**F**) CCD similarity score matrix of proteins in (**D**) ATM clique, (**E**) FA clique and (**F**) mixed clique. **G**–**I** Similarity scores of proteins in (**G**) ATM clique, (**H**) FA clique and (**I**) mixed clique in K562, RPE-1 p53$^{KO}$ and p53/ BRCA1$^{dKO}$ cells. **J**–**N** M- and N-synergies discussed in the text. Column labels are names of proteins or the inhibitor used ('i' suffix). Proteins or inhibitors with significant CCDs (FDR$_{CCD}$ < 0.05) are marked with an asterisk. Chromatin features are colored as in Fig. 1C. **J** ATM signaling. **K** Fanconi anemia complex. **L** SMC5/6 complex. **M** DNAPK$_{cs}$ KO and inhibition. **N** BRCA1-A complex. Source data are provided as a Source Data file.

M-synergy although it did not pass the significance threshold (Fig. 4L). These data indicate that the SMC5/6 complex preferentially modulates MMEJ:NHEJ balance in H3K27me3 and lamina-associated heterochromatin.

**Role of ATM signaling in heterochromatin**
To further investigate the CCD of ATM in heterochromatin, we treated cells with the ATM kinase activity inhibitor KU55933 (Supplementary Fig. 9A, B). ATM inhibition exhibited significant M-synergies with H3K27me3 and interactions with LMNB1, but did not exhibit M-synergies with other triple heterochromatin features. This CCD pattern is more similar to CHEK2, ATM's main signal transducer[34], than ATM itself (Fig. 4J). This suggests that loss of ATM downstream signaling impacts CCDs differently than losing ATM itself, in line with earlier observations that loss and inhibition of ATM can have different

effects[35,36]. These data underscore the importance of the ATM signaling axis in repair of DSB in heterochromatin.

**BRCA1 regulates ATM global effect but not detectably CCD**
In RPE-1 cells we noticed that the global effect of ATM, which favors MMEJ, is weaker when BRCA1 is absent (Fig. 5A; see also Fig. 3B). This was also observed upon KU55933 inhibition of ATM (Fig. 5B). In contrast, BRCA1 did not detectably affect the heterochromatin CCD of ATM (Fig. 5C, D). Although the statistical power of this analysis was somewhat limited, this suggests that the global and chromatin-dependent cross-talk between ATM and BRCA1 involve distinct mechanisms. Possibly, ATM regulates MMEJ:NHEJ balance by phosphorylation of multiple targets, BRCA1 being one of them[37]. However, the M-synergy of ATM with heterochromatin may be driven by other protein targets and does not depend on BRCA1.

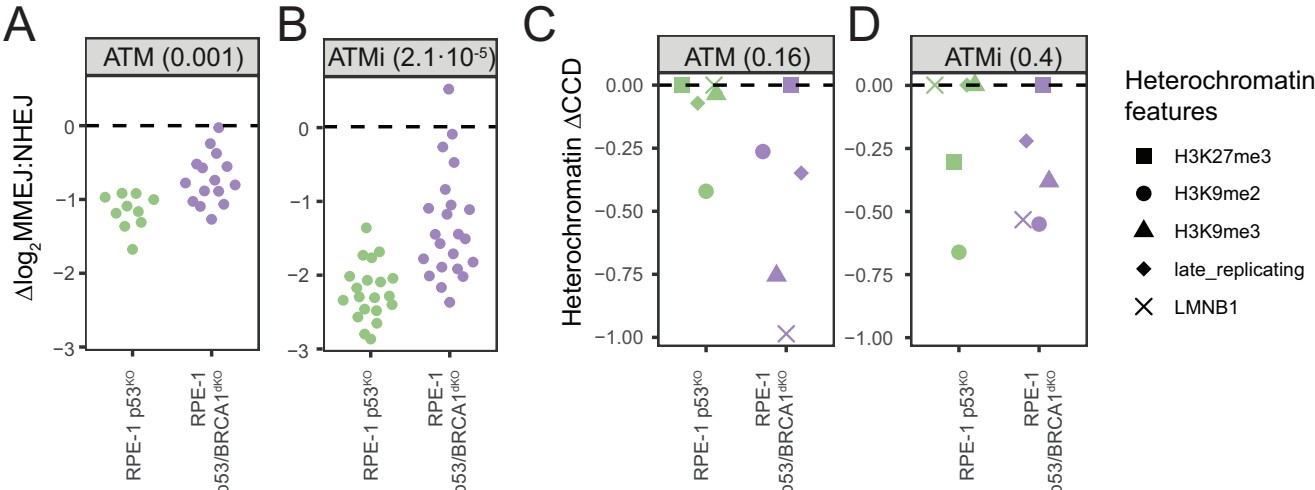

**Fig. 5 | CCD of ATM signaling in heterochromatin is independent on BRCA1.**
**A**, **B** Global effect measured as $\Delta log_2$MMEJ:NHEJ scores in RPE-1 p53$^{KO}$ and p53/BRCA1$^{dKO}$ cells after (**A**) ATM KO and (**B**) ATM inhibition. *P*-value shown in parentheses are according to two-sided Student's *t*-test. **C**, **D** Heterochromatin CCD effect measured as the estimated $\Delta log_2$MMEJ:NHEJ for every heterochromatic feature after (**C**) ATM KO and (**D**) ATM inhibition. *P*-value shown in parentheses are according to two-sided paired Student's *t*-test. Source data are provided as a Source Data file.

## Highlights of N-synergies

Among canonical components of the NHEJ pathway, only POLL exhibits significant N-synergy. Our data indicate that the ability of POLL to promote NHEJ is facilitated by euchromatin, particularly in transcribed regions. DNA-PKcs (PRKDC), another crucial regulator of NHEJ, showed only a weak, non-significant N-synergy pattern. However, treatment of cells with the DNA-PKcs inhibitor M3814 yielded a N-synergy pattern that was similar but much stronger (Fig. 4M, Supplementary Fig. 9A, B). The consistent pattern indicates that DNA-PKcs is primarily N-synergistic with transcribed parts of the genome, and to a lesser extent with triple heterochromatin. Other N-synergistic proteins have previously been linked to various other repair pathways, underscoring extensive cross-talk between pathways[38]. An example is the BRCA1-A complex, which fine tunes BRCA1-mediated resection[39,40]. Its subunits BRCC36 (BRCC3), RAP80 (UIMC1) and BRE (BABAM2) exhibit N-synergies with euchromatic features, and ABRAXAS1 shows similar patterns but did not pass the significance threshold (Fig. 4N). Furthermore, BRCA2 shows N-synergy with euchromatin (Supplementary Fig. 4A). In this case the N-synergy may be due to local suppression of MMEJ, because suppression of MMEJ by BRCA2 has been reported[41,42]. However, we do not detect these CCD patterns in RPE-1 cells. We conclude that the cross-talk between pathways might be different between cell lines and that could potentially explain the different results in K562 and RPE-1 cells.

## Testing impact of CCDs on human cancer genomes

Several of the hits in our CCD screen are frequently mutated in human cancers. We hypothesized that this would affect the genome-wide distribution of mutations that are generated by MMEJ and NHEJ. To test this, we analyzed genomes from a diversity of tumor types with driver mutations in either ATM, MEN1, SETD2, BRCA1, BRCA2 and ATRX. We chose these drivers based on the availability of sufficient numbers of sequenced cancer genomes carrying these driver biallelic deletions (with no mutations in other repair proteins) as well as did not show mutational signatures associated with other DNA repair deficiencies (see Methods)[43,44] (Supplementary Fig. 10A). In genomes from each driver-tumor type combination, we then scored signature short deletions that are characteristic of MMEJ and NHEJ[45]; determined the pathway balance in euchromatin (constitutive inter-LADs) and lamina-associated heterochromatin (constitutive LADs); and calculated the

fold difference between these two chromatin contexts. Next, we compared this $log_2$ fold difference to that of a set of tissue-matched control tumors (see Methods).

## CCDs in tumors generally match screen results

Based on the observed M-synergy of ATM, MEN1, SETD2 and BRCA1 in triple heterochromatin (for BRCA1 only detected in the separate analysis (Supplementary Fig. 7D)), we predicted that in tumors that are homologous mutant for these proteins—compared to tumors with the respective wild-type protein being present—the $log_2$MMEJ:NHEJ ratio should decrease relatively more in heterochromatin compared to euchromatin. Indeed, this is what we observed for ATM$^{-/-}$ and MEN1$^{-/-}$ tumors (Fig. 6A, B). In BRCA1$^{-/-}$ tumors the median differences are statistically significant but more subtle (Fig. 6C), with tumors from two out of three tissues tested showing the predicted effect orientation (Supplementary Fig. 10C). SETD2$^{-/-}$ tumors showed only non-significant effects, although the direction of the trend was as predicted by the screen (Supplementary Fig. 10C). Because SETD2 is also a regulator of chromatin, it is possible that its loss in tumors alters the chromatin landscape to such a degree that our assumption of invariant coordinates of constitutive LADs and iLADs is incorrect. In tumors lacking ATRX (N-synergy in triple heterochromatin), we expected the $log_2$MMEJ:NHEJ ratio to increase relatively more in heterochromatin compared to euchromatin. Despite this difference being small, this is what we observe (Fig. 6D). Finally, we predicted that in BRCA2$^{-/-}$ tumors the $log_2$MMEJ:NHEJ ratio should increase relatively more in euchromatin compared to lamina-associated heterochromatin (Supplementary Fig. 4A). Indeed, this is what we observed (Fig. 6E).

## Further analysis of BRCA2$^{-/-}$ tumors confirms CCDs

Unlike the other tumor cohorts, BRCA2$^{-/-}$ tumors have 7-fold higher short deletions compared to their controls (Supplementary Fig. 10D), which might confound the results. We therefore also compared a cohort of BRCA2$^{-/-}$ tumors to a cohort of BRCA2$^{+/+}$ genome-instable head and neck squamous cell carcinoma ((HNSCC) cohort[46]; Supplementary Data 7; see Methods). Although these HNSCC tumors still do not meet the rate of short deletions found in BRCA2$^{-/-}$ tumors, the difference is less (Supplementary Fig. 10E). Importantly, the detected CCD in this comparison was even stronger (Fig. 6E, F), indicating that the overall mutation rate is unlikely to be a confounding factor.

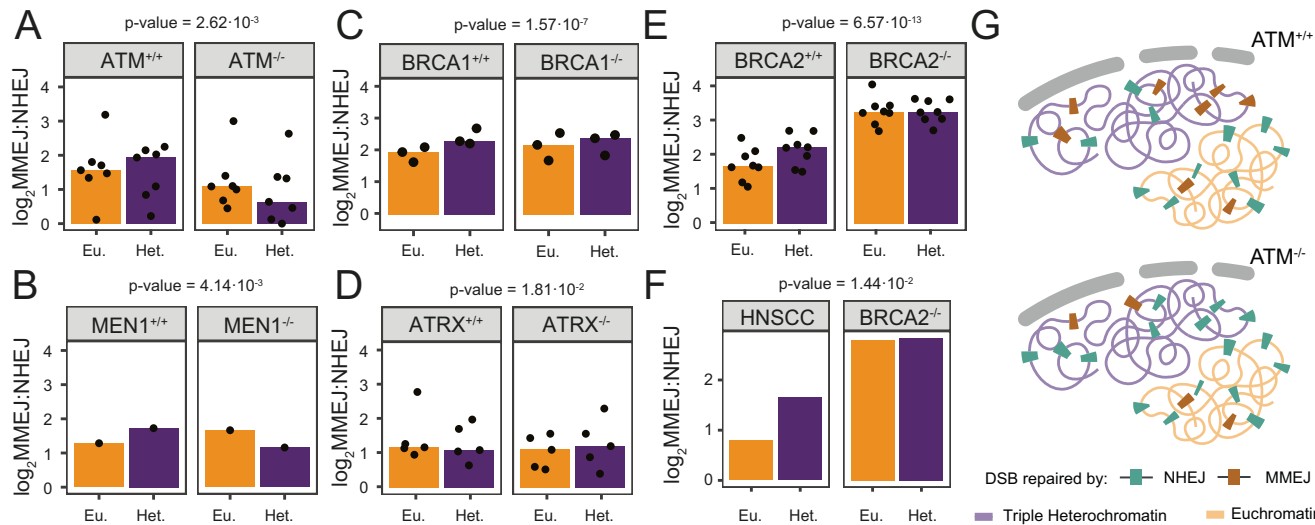

**Fig. 6 | *Impact on mutation distribution in cancer genomes.* A–E** Total MMEJ and NHEJ signature indel log$_2$ ratio in euchromatin (Eu.) and constitutive lamina-associated heterochromatin (Het.) in genomes of tumors with indicated genotypes. Control tumors ($^{+/+}$) were tissue-matched as much as possible to the respective mutant tumors ($^{-/-}$) (Supplementary Fig. 10A). Black dots represent values of individual tumor subtypes and the bars represent the median value across tissues and two-sided *p*-values are shown in the figure. **F** same as **A**–**E**, but for BRCA2$^{-/-}$ tumors compared to genome-instable HPV$_{neg}$ HNSCC, which have a more similar overall mutation rate than the set in (**E**). **G** Cartoon illustrating ATM M-synergy with triple heterochromatin. In ATM$^{+/+}$ tumors (topleft panel), triple heterochromatin has a higher abundance of MMEJ mutations (brown speckles) compared to NHEJ mutations (green speckles), relative to euchromatin. In ATM$^{-/-}$ tumors (right panel), euchromatin remains relatively unchanged, while triple heterochromatin shows a stronger reduction of MMEJ mutations than of NHEJ mutations. Source data are provided as a Source Data file.

Interestingly, large deletions (>1.4 kb) with microhomologies at their break sites (which we assume to be primarily repaired by MMEJ) also showed a striking shift towards euchromatin in BRCA2$^{-/-}$ tumors compared to BRCA2$^{+/+}$ HPV$_{neg}$ HNSCC (Supplementary Fig. 10F), consistent with N-synergy of BRCA2 with euchromatin. These additional analyses underscore that loss of BRCA2 causes a shift in repair-induced mutations dependent on the chromatin context.

### Overall concordance between K562 screen and cancer data

Taken together, this survey indicates that—on average across the included tumor types—loss of five of the six tested proteins significantly alters the MMEJ:NHEJ balance between heterochromatin and heterochromatin, with the direction of the effect being concordant with the CCDs as detected in our K562 screen (Supplementary Fig. 10C). We note, however, that not every individual tumor type showed concordant results with the CCDs in K562 cells (Supplementary Fig. 10C). This may be due to strongly altered cLAD-ciLAD landscapes in certain tissues, or due to cell-type specific modulation of CCDs.

## Discussion

An important implication of our findings is that the interplay between chromatin context and the DSB repair machinery may be much more widespread and complex than previously appreciated[47]. Most likely, other DNA repair pathways that we did not probe are also subject to CCDs[48]. Rather than the current view that only a small number of repair proteins sense certain specific chromatin types, we suggest that the interplay with chromatin involves a division-of-labor among dozens of repair proteins. This complexity seems akin to the regulation of human gene expression: most genes are regulated by a large number of proteins, each contributing a bit to the overall expression level. This complexity poses challenges to the dissection of the underlying mechanisms by traditional reductionist molecular biology approaches. Additional high-throughput approaches combined with advanced computational modeling may help to address this challenge.

Generally, the effect sizes of CCDs of individual proteins that we uncovered here are modest[49] modest (typically do not exceed ~50%,

see Supplementary Methods, Supplementary Fig. 6E). However, penetrance of the KOs in our K562 screen is incomplete, causing underestimation of the effect sizes (see Supplementary Fig. 7 for multiple lines of evidence). Moreover, considering the large number of proteins exhibiting CCDs, their collective effect is likely to be much more substantial than that of individual proteins.

Our data demonstrate that CCDs occur in multiple cell types, and are broadly similar between K562, RPE-1 and tumor cells. However, further analysis is needed, as the chromatin preferences of some repair proteins are not identical between these cell types. This may be due to differences in chromatin composition, or due to inaccuracies in the chromatin maps that we used. It is also possible that some repair proteins act differently in certain cell types, e.g. due to posttranslational modifications or different partner proteins. Furthermore, the K562 and RPE-1 cell lines that we employed lack TP53, a major player in the DNA damage response.

The molecular mechanisms underlying the observed CCDs are likely to be highly diverse. Two broad categories may be distinguished. In the first category, the ability of a repair protein to tune the MMEJ:NHEJ balance is modulated by physical interactions with a chromatin protein that marks a specific chromatin state. In the second category, the repair protein doubles as a component of a particular chromatin state, and its depletion causes a change in that chromatin state, which in turn is detected by other components of the repair machinery. Examples of the latter category may be SETD2, SMC proteins, INO80, NIPBL, TRRAP and CHAF1A, and possibly BRCA2 which has been reported to regulate transcription[50]. Most DSB repair proteins, however, do not appear to be stable components of chromatin but are instead thought to be recruited locally to DSBs[51]. We thus expect that most of the underlying mechanisms will be of the first category.

We note that the repair of Cas9-induced DSBs is likely to be different from repair of breaks by other sources, and hence caution should be taken when interpolating our findings to other types of DSBs. However, our results indicate that CCDs also play a role in the genome-wide distribution of accumulated mutations in a diversity of cancer types, which are triggered by a diversity of other DSB-inducing

agents. While these findings do not immediately lead to therapeutic applications, they help to better understand how the patterns of mutations in cancer genomes arise. Further studies that expand these relationships to other indel and structural variants could help to gain further insight into genome evolution in DNA repair deficient tumors.

Possibly, CCDs could be exploited to enhance gene editing efficiency in chromatin contexts that are difficult to edit. We have recently analyzed the CCD of epigenetic drugs and identified several drugs that can improve gene editing efficiency in particular chromatin contexts[49]. Potentially, combinations of drugs targeting a specific DNA repair protein as well as a specific chromatin feature could further improve this strategy, but this will require systematic testing.

# Methods

## Cell line and culture conditions

We used the clonal cell line K562#17 DSB-TRIP clone 5[6], which is a genetically modified monoclonal human K562 cell line (ATCC). This cell line stably expresses Shield1-inducible DD-Cas9 and additionally carries 19 uniquely barcoded integrated pathway reporters (IPRs) in precisely mapped genomic locations (Supplementary Data 4). Cells were cultured in RPMI 1640 (1187-093 GIBCO) supplemented with 10% fetal bovine serum (FBS, Capricorn Scientific) and 1% penicillin/streptomycin (15070-063 GIBCO). For this study, we also generated DSB-TRIP cell pools derived from RPE-1 p53$^{KO}$ and RPE-1 p53/BRCA1$^{dKO}$ cells that constitutively express Cas9[24]. All RPE-1 derived cell lines (kindly shared by Jonkers lab at the NKI) were cultured in DMEM/F12 (1:1) medium (11320-033 GIBCO) supplemented with 10% fetal bovine serum and 1% penicillin/streptomycin at 37 °C at 5% $CO_2$. Cells were regularly checked to be free of mycoplasma.

## Design of KO gRNA library

We designed an arrayed CRISPR/Cas9 KO gRNA library (KO gRNA library, in short) which targeted a total of 519 genes encoding proteins previously linked to DNA repair. The list of proteins was based on the Gene Ontology term GO:0006302 (double strand break repair), supplemented with a manually curated list. The crRNA library was generated by Integrated DNA Technologies (IDT) and contained 4 crRNAs per gene (Supplementary Data 1). The individual crRNAs were delivered in a lyophilized RNA form and were diluted in Duplex Buffer (DB, IDT cat. no. 11-01-03-01) to a stock concentration of 100 µM. Finally, we pooled crRNA targeting the same gene to a single well in a final concentration of 5 µM per crRNA.

## Screen procedure

We performed the semi-automated arrayed screen in 96-well format. It consisted of the following key steps:

- Day 1: Induction of Cas9 expression and transfection with gRNAs to disrupt 519 individual genes.
- Day 5: Passaging of cells; quality checks of liquid handling and transfection efficiency (scheme 2)
- Day 6: Second transfection: induction of DSBs in the IPRs.
- Day 9: Lysis of cells.
- Downstream processing: PCR amplification and sequencing of the barcoded IPRs.

A detailed procedure can be found in the Supplementary Methods section.

## Chromatin context effects in RPE-1 cells

To test if chromatin context-dependent effects occur in other cells than K562, we generated DSB-TRIP pools RPE-1 p53$^{KO}$ and RPE-1 p53/BRCA1$^{dKO}$[24]. Note that—in contrast to K562 clone 5—these cell pools are not clonal lines but a mixture of clones that each carry 1.56 and 0.26 IPRs/cell on average. We generated TRIP pools with fewer IPRs per cell than usual[52] to avoid cell-cycle arrest caused by an excess of DSBs per

cell[53]. In these cell lines, we knocked out 20 DNA repair proteins with chromatin-context effects.

## DSB-TRIP pool generation

We generated DSB-TRIP pools were as described in ref. 49. In brief, we transfected RPE-1 cells with pPTK-BC-IPRv2 and PG transposase-mCherry plasmids and sorted for mCherry positive cells. A week after transposase inductions, we sorted 250 p53$^{KO}$ and 1000 p53/BRCA1$^{dKO}$ mCherry negative RPE-1 cells. Then, we mapped IPR integrations sites by inverse PCR. These pools contained 261 and 183 mapped IPRs respectively with an estimation of 1.56 and 0.26 IPRs/cell.

## Lentiviral gRNA pool design and production

To test if DNA repair proteins have CCDs in RPE-1 cells, we designed gRNA pools targeting 20 DNA repair proteins that showed CCDs in K562 cells. These include 4 proteins that (in K562) *favor NHEJ* (POLL, BRCA2, CHAF1A and BRCC3), 12 protein that *favor MMEJ* in heterochromatin (ATM, ATR, CHEK2, FAAP24, FANCD2, FANCG, FANCM, MDC1, RAD50, RBBP8, SMC5 and TOPBP1), 3 proteins that *favor MMEJ* in euchromatin (BLM, PARP1 and RMI2) and 1 protein that *favor NHEJ or MMEJ* depending on the chromatin context (BOD1L1).

We designed up to 17 gRNAs targeting the first exons without any off-target predicted with at least 2 mismatches targeting a window of maximum 4000 bp. gRNA pools were designed using IndePhi (https://indelphi.giffordlab.mit.edu/about)[54] and mismatch analysis was performed using CRISPRoff webtool (https://rth.dk/resources/crispr/crisproff/)[55] and ordered as oPools (IDT, 50 pmol/oligo, Supplementary Data 1). Next, we assembled 50 nM ssDNA oligo pools with 50 ng pLenti-gRNA-mCherry-Puro plasmid[56] using NEBuilder HiFi DNA assembly master mix (New England Biolabs, cat. no. E2621) and transformed 2 µl into NEB 5-alpha competent bacteria (New England Biolabs, cat. no. C2987H). We grew the transformation mix in LB medium and purified each plasmid pool with Purelink HiPure Plasmid Midiprep kit (ThermoFisher, cat. no. K210005).

We produced one viral supernatant for each pool of gRNAs targeting a single gene (total of 20 preparations) and one viral supernatant with a non-targeting control gRNA. These lentiviruses were generated by contransfecting 6 µg pool of gRNAs with 1.5 µg pMD2G (Addgene #12259) and 4.5 µg psPAX2 (Addgene #12260) packaging plasmids in HEK293T cells. Medium was changed 6 h post transfection and we harvested lentivirus containing medium 24 and 48 h post transfection.

## Transient KO generation and pathway balance assay

To test the chromatin-effects of DNA repair proteins, we transiently knocked-out each of the 20 repair proteins.

First, we reverse transduced 100,000 cells in 1.5 ml medium containing 10 µg/ml polybrene on 250 µl of each lentiviral supernatant. As a control, we included a non-transduced sample that we used to control for proper puromycin selection. Thirty-six hours post transduction, cells were inspected for mCherry positivity, and we replaced the culture medium with medium containing 10 µg/ml puromycin. Puromycin concentration was defined previously as the minimum concentration for efficient killing of untransduced cell-lines. Cells were cultured with puromycin containing medium for 4 days.

After puromycin selection, cells were replated in 6-well plates at different densities for the pathway balance assay. We plated 100,000 RPE-1 p53$^{KO}$ and 120,000 RPE-1 p53/BRCA1$^{dKO}$ cells per well. These concentrations were previously optimized to achieve 90% confluency at the end of the assay. 24 h after re-plating the cells, we transfected with LBR2 gRNA that created the DSB at the reporters. To do so, we mixed 20 nM of LBR2 crRNA with 20 nM tracrRNA in 25 µl of Duplex Buffer and diluted 1:500 RNAiMAX lipofectamine with 175 µl Optimem. After 5 min incubation time, we mixed LBR2 crRNA::tracrRNA with the diluted lipofectamine and incubated the mix for 15 min. Then, we added 200 µl of the transfection mix dropwise on the cells. To avoid

any toxicity caused by lipofectamine, we replaced the transfection medium by regular medium 10 h after the transfection.

## Indel library preparation

Seventy-two hours post-transfection, we harvested and lysed the cells in 50 μl DirectPCR Lysis Buffer for 16 h at 55 °C and inactivated at 85 °C for 45 min. After lysis, we quantified the amount of DNA in the lysates by Qubit DNA dsHS Assay Kit to account for lower cell titers in the harvested samples because of toxicity of the KO. Then, we performed the indelPCRs as described previously for the DSB-TRIP pools, with the following modifications. We performed the indelPCR in triplicate to capture a higher proportion of the complexity in the sample and starting lysate amount varied from cell line to cell line. We used the same index combinations for all three replicates and the sample was sequenced on a NextSeq 550 with 150 bp long single reads.

## Data analysis

We analyzed the CCDs in RPE-1 cells as performed for the K562 screen data (see data processing section), with slight modifications. First, the indel scoring metadata file was updated to account for extra 8 nucleotides present in pPTK-BC-IPRv2 reporter compared to the original one. Secondly, we filtered IPRs based on IPR frequency (0.0025 in p53$^{KO}$ cells and 0.0075 p53/BRCA1$^{dKO}$ cells). An IPR was only considered for the CCD analysis if we found it in three replicates. After this processing, we ran the CCD analysis pipeline in an average of 10.6 IPRs in RPE-1 p53$^{KO}$ and 15.4 IPRs in RPE-1 p53/BRCA1$^{dKO}$ cells (Supplementary Fig. 8B). We ran the CCD analysis using publicly available 10 chromatin tracks of RPE-1 cells (Supplementary Data 3).

## Chromatin context effects assessed with inhibitors

For DNAPK and ATM we also determined CCDs in K562 cells by using specific small-molecule inhibitors. The experimental design was similar to the KO screen setup, with the following modifications:

## gRNA transfection

To induce DSBs, we introduced LBR2 gRNA into K562 cells by plasmid nucleofection instead of RNA transfection. For this purpose, we resuspended one million K562 clone 5 cells in 100 μl transfection buffer (100 mM KH2PO4, 15 mM NaHCO3, 12 mM MgCl2, 8 mM ATP, 2 mM glucose (pH 7.4))[57]. Then, we added 12 μg of either gRNA-containing LBR2 plasmid or GFP-expressing control plasmid. Cells were electroporated in an Amaxa 2D Nucleofector (T-016 program). Twenty-four hours post-nucleofection, we assessed transfection efficiency by visual observation of GFP-positive cells. This GFP sample was later used as non-targeted control. In RPE-1 cells, we followed the protocol described previously.

## Inhibitor treatment

Eight hours after nucleofection, we added 500 nM Shield-1 (Aobious) to stabilize DD-Cas9 protein. Together with Shield-1, we added inhibitors of either DNAPK (M3814, final concentration 1 μM from a 1 mM stock in DMSO, MCE cat. no. HY-101570), ATM (KU5593, final concentration 10 μM from a 10 mM stock in DMSO, Calbiochem cat. no. #118500), and DMSO-only vehicle controls (1:1000, Sigma cat no. D4540). In RPE-1 cells, we added 1 μM Shield-1 and 1 mM Doxycycline 24 h prior to the gRNA transfection. At the same time of the transfection, we added ATM (KU5593, final concentration 10 μM from a 10 mM stock in DMSO, Calbiochem cat. no. #118500), and DMSO-only vehicle controls (1:1000, Sigma cat. no. D4540).

## Indel library preparation

Seventy-two hours after DD-Cas9 stabilization, we harvested the cells, performed genomic DNA (gDNA) extraction with the ISOLATE II genomic DNA kit (Bioline, BIO-52067) and diluted DNA to 50 ng/μl. Indel sequencing libraries were prepared as described for the screen

but with minor changes as follows. We performed indelPCR1 with 200 ng of gDNA as input (4 μl of 50 ng/μl concentrated sample) and 200 nM of each primer for 4 cold cycles and 8 hot cycles. Then, we performed indelPCR2 with 5 μl indelPCR1 product and 166.6 nM of each primer for 1 cold cycle and 13 hot cycles. We pipetted both PCR reactions manually. We pooled samples in equimolar ratios and prepared them for sequencing as described for the screen. Samples were sequenced on a MiSeq with 150 bp single-end reads and including 10% of PhiX spike-in. We performed this experiment in three independent biological replicates. We performed RPE-1 libraries following the same protocol as in K562 cells with the following exceptions. In indelPCR1, we used 500 nM of each primer for 5 cold and 8 hot cycles. Samples were sequenced on a NextSeq 550 with 150 bp single-end reads and including 15% of PhiX spike-in.

## Processing and statistical analysis of K562 screen data

We processed and analyzed the K562 screen data a workflow with the following key steps:

1. Demultiplexing and general quality control of sequencing reads.
2. Scoring of indels in IPRs.
3. Calculation of changes in MMEJ:NHEJ balance.
4. Identification of proteins with global effects on MMEJ:NHEJ balance.
5. Identification of proteins with CCD: three-step linear modeling.
   a. Initial selection of proteins with any effect on MMEJ:NHEJ balance.
   b. Principal component regression.
   c. Linear modeling to identify individual protein−chromatin feature links.
6. Estimation of chromatin context dependent MMEJ:NHEJ balance changes.
7. Estimation of screen KO penetrance.
8. Data visualization.

A detailed explanation of each of these key steps is available in the Supplementary Methods.

## CCD analysis on protein knock-outs in RPE-1 pools

We analyzed the CCDs in RPE-1 cells as performed for the K562 screen data (see data processing section below), with slight modifications. First, the indel scoring metadata file was updated to account for extra 8 nucleotides present in pPTK-BC-IPRv2 reporter compared to the original one. Second, we filtered IPRs based on IPR frequency (0.0025 in p53$^{KO}$ cells and 0.0075 p53/BRCA1$^{dKO}$ cells). An IPR was only considered for the CCD analysis if we found it in three replicates. After this processing, we ran the CCD analysis pipeline in an average of 10.6 IPRs in RPE-1 p53$^{KO}$ and 14.6 IPRs in RPE-1 p53/BRCA1$^{dKO}$ cells (Supplementary Fig. 7B). We ran the CCD analysis using publicly available 10 chromatin tracks of RPE-1 cells (Supplementary Data 3) processed as described in[58]. A list of integration coordinates and chromatin features of all the IPRs assayed in RPE-1 cells can be found in Supplementary Data 4.

## CCD analysis on inhibitor experiments (K562 and RPE-1)

We analyzed the CCDs of inhibitors as performed for the screen data (see data processing section below), with slight modifications. We tested the significance of the perturbation by means of a Student's t-test instead of a z-test. We used this test because here each replicate includes only a single control sample. Everything else was performed as described for the screen data. For the RPE-1 inhibitor experiment, we processed the data as described for RPE-1 pools with the exception that only IPRs with 40 NHEJ and MMEJ reads were used.

## Proteins with CCD classified by Gene Ontology terms

We classified the 89 proteins with CCDs by their Gene Ontology (GO) terms. For this analysis, we used five GO terms that are under the DSB

repair (GO:0006302): DSB repair via non-homologous end joining (GO:0006303), DSB processing (GO:0000729), DSB repair involved in meiotic recombination (GO:1990918), DSB repair via single-strand annealing (GO:0045002) and DSB repair by homologous recombination (GO:0000724). We excluded the other categories either because no protein in the category had CCDs (mitochondrial DSB repair) or the terms were ambiguous (positive and negative regulation of DSB repair). We renamed the GO:0006303 term to DSB repair by end-joining pathways to avoid confusion with NHEJ. As stated in the description of the GO term, this term includes proteins that are related to NHEJ and/or MMEJ. Finally, we retrieved the list of human proteins annotated in these categories and visualize the ones with significant CCDs as a heatmap.

### Comparison to protein-protein interaction data

To assess if physically interacting proteins tend to have a similar CCDs, we computed cosine similarities of synergy scores (Eq. 1) between physically interacting protein pairs and compared them to the synergy scores expected by random chance. First, we computed the cosine similarity matrix for all proteins with significant CCDs with the *lsa* package (version 0.73.3). For this we compared the 25 synergy scores for each protein. We decided to use the cosine distance as a similarity score over other metrics, because it deals best with data containing zero values. Second, we selected protein pairs that physically interact in living cells according to the BioGrid database (release version 4.4.209)[29]. A total of 118 physical interactions were reported between proteins in our dataset. These interactions were detected with one of the following methods as reported by BioGrid database: Affinity Capture-MS, Affinity Capture-Western, Co-localization, Co-crystal structure, Co-purification, Co-fractionation, FRET, PCA, proximity label-MS and Two-hybrid. To determine whether the average cosine distance of the 118 interacting protein pairs was significantly different from that of random pairs of proteins, we compared it to the distribution of mean cosine distances obtained from 1000 randomly selected sets of 118 protein pairs.

$$\cos(A,B) = \frac{AB}{||A||\,||B||} = \frac{\sum_{i=1}^{n} A_i B_i}{\sqrt{\sum_{i=1}^{n} A_i^2}\,\sqrt{\sum_{i=1}^{n} B_i^2}} \tag{1}$$

*Where A and B are a vector of 25 synergy scores for a protein.*

We also explored if proteins forming interaction cliques tend to have similar CCDs. To do so, we built an interaction network of physical interactions using the *igraph* package (version 1.3.4) and identified highest order cliques. We found three cliques with four elements each and displayed them on the UMAP plot (Fig. 4C).

### CCD pattern similarity in K562 and RPE-1 cell lines

To assess if DNA repair proteins have similar CCD patterns in K562 and RPE-1 cells, we computed cosine similarities using the formula described before for each RPE-1 cell line separately. Due to the limited number of sampled IPRs in RPE-1 cells, we only compared synergy scores of chromatin features that we sampled sufficiently. We considered that a chromatin feature is sufficiently sampled when we sampled at least 2 IPRs that are embedded in that chromatin feature (Z-score > 0.5). In RPE-1 p53/BRCA1$^{dKO}$ cells, we compared the 10 synergy scores for each protein (Supplementary Data 5). In RPE-1 p53$^{KO}$, we excluded synergy scores with LMNB1, H3K27me3, H3K9me3 and H3K4me3 chromatin features from this analysis and compared 6 synergy scores for each protein. To test whether the average cosine similarity between cell types was higher than expected by chance, we selected 18 random proteins in the screen and calculated the cosine similarity scores between their K562 pattern and the pattern of one of the proteins tested in RPE-1 cells. We repeated this calculation

1000 times, calculated the average distribution and compared it to the similarity scores of the matching protein pairs.

### Chromatin context dependent pathway activity in tumors

The full list of driver mutations identified by the pan-cancer analysis of whole genomes (PCAWG) consortium was obtained for all available cancer subtypes (https://dcc.icgc.org/releases/PCAWG/driver_mutations)[43,44]. Driver events (i.e., substitutions, indels or copy number alterations) were identified in genes of interest (GOI) encoding for 21 proteins with significant CCDs in K562 cells: *ATM, ATR, ATRX, BLM, CHEK2, EGFR, FANCD2, FANCF, FANCM, HELQ, INO80E, L3MBTL2, MEN1, PARG, RAD17, SETD2, SMC5, TRRAP, USP* and *XRCC1* as well as cases of full *BRCA1* and *BRCA2*-deficiency. We selected genes among GOIs with at least 5 cancer samples with a loss-of-function (LOF) driver mutation of which 3 belong to the same cancer subtype. Genes that passed this filter included *ATM, ATR, ATRX, BLM, BRCA1, BRCA2, CHEK2, MEN1* and *SETD2*.

We considered a cancer sample with a GOI driver mutation as a case ("mutant") when: (I) the cancer sample carried a LOF driver mutation in only 1 of the 22 GOIs; (II) did not exhibit markers of a DNA repair deficiency−based on PCAWG BRCA1/2 deficiency status (except for the *BRCA1/2*-deficiency groups), PCAWG mismatch repair deficiency status, MUTYH mutational signature (COSMIC SBS36) and POLE/POLD1 proofreading/exonuclease-deficiencies (COSMIC SBS10a, SBS10b, SBS10c, SBS10d and SBS20); (III) did not display mutational signatures indicative of previous mutagenic treatments or other very rare mutational signatures; and (IV) cancer samples had at least 500 substitutions genome-wide−as an indirect measurement for low sequencing quality, low coverage, low tumor purity and to exclude cancer subtypes with overall very low total mutation burdens (e.g., pediatric brain cancer for *SETD2*). We considered a cancer sample as a control when it fulfilled conditions II, III and IV, but did not have a driver mutation, LOF or otherwise, in any of the 22 GOIs.

Indels were obtained from the final PCAWG consortium somatic mutation list for all cases and controls. The methods and post-calling indel filtering strategies were previously described in detail[45]. Indels were subsequently classified using *indelsClassification* (https://github.com/ferrannadeu/indelsClassification) to identify deletions generated by error-prone NHEJ (>5 bp deletions without micro-homology) and MMEJ repair (>5 bp deletions with ≥2 bp micro-homology sequence). Taken together, for each of the 6 GOIs we have at least 3 cancer samples carrying a GOI driver mutation in at least one cancer subtype. These are matched by a larger group control cancer sample. PCAWG, has for some cancer subtypes (e.g., breast cancer), multiple datasets predominantly defined by the continent/country of primary sample collection (Supplementary Fig. 10A). We opted to keep these datasets as separate entities (e.g., cancer subtypes), because we observed small, but detectable, differences in the MMEJ:NHEJ balance between datasets of the same cancer subtype based on the post-selection control samples (Supplementary Fig. 10B).

Next, we determined the number of MMEJ and NHEJ indel events in heterochromatin and euchromatin regions for each mutant and control sample. Because it was not feasible to map heterochromatin and euchromatin in this broad diversity of tumors, we assumed that previously defined ciLADs (constitutive lamina-associated domains) and ciLADs (constitutive inter-LAD regions), which are strictly conserved across nine different cell types, would be a reasonable approximation of heterochromatin and euchromatin domains, respectively[59]. Coordinates were downloaded from https://osf.io/dk8pm/.

Barring DNA repair deficiencies, MMEJ or NHEJ-generated indel mutations are sparse genome-wide in a number of the selected cancer subtypes. Assuming that case and control cancer samples are similar in indel composition we calculate the number of MMEJ or NHEJ deletion events per cLAD or ciLAD region per GOI by summing over the cases or

controls (Eq. 2). Lastly, we calculated the log$_2$MMEJ:NHEJ balance in cLADs and ciLADs per each GOI−cancer subtype combination in mutant and controls samples.

$$MMEJ_{cLAD,GOI,cancer\ subtype,cases} = \sum_{sample\ \in\ (GOI\ \cap\ cancer\ subtype\ \cap\ cases)} MMEJ_{cLAD,sample}$$

(2)

To assess whether a GOI driver mutation has an impact on genome-wide distribution of MMEJ and NHEJ mutations per cancer subtype, we calculated the pathway balance log$_2$ fold-change between cLADs and ciLADs (Δlog$_2$MMEJ:NHEJ$_{Het/Eu}$). For this analysis, we assume: (I) that cases with a driver mutation are similar in indel composition, and (II) that there is some heterogeneity in the indel composition of control cancer samples due to the far larger sample size. To get a better assessment of the possible heterogeneity in Δlog$_2$MMEJ:NHEJ$_{Het/Eu}$ for the controls per GOI−cancer subtype combination, we performed bootstrapping[60]. By sampling the same number of control samples with replacement among the control set for 1000 bootstrap samples, and assessing the sampling distribution of the Δlog$_2$MMEJ:NHEJ$_{Het/Eu}$ based on 1000 samples. Using the mean and the standard deviation of the sampling distribution, we Z-score transformed the Δlog$_2$MMEJ:NHEJ$_{Het/Eu}$ of the mutant samples for each GOI-tumor subtype combination. With the exception of *MEN1*, all GOIs have 2 or more cancer subtypes. To get a general assessment whether the GOI overall disbalances the MMEJ-NHEJ ratios in cLAD and ciLAD regions, we combine the individual cancer subtype Z-scores per GOI using Stouffer's Z-score method (Eq. 3).

$$Z_{GOI} = \frac{\sum_{cancer\ subtype} Z_{GOI,cancer\ subtype,cases}}{\sqrt{k}}$$

(3)

Lastly, we transformed combined Z-scores into *p*-values and corrected them for multiple testing using Benjamini-Hochberg methods. We concluded that a GOI affects the genomic distribution of NHEJ and MMEJ short deletions, when $p_{adj} < 0.05$. Using this method we assess the overall impact of a GOI driver mutation on the MMEJ:NHEJ balance in cLAD and ciLAD regions corrected for background differences between cancer subtypes and datasets, and taking into account potential heterogeneity within the control samples.

### Comparing BRCA2-deficient to BRCA2-proficient cancers

Due to the high mutational load of BRCA2-deficient compared to BRCA2-proficient cancers, we decided to check if the CCD pattern of BRCA2 was also present compared to genome-instable tumors. For this reason we chose a recent whole-genome sequencing dataset derived from HPV negative head and neck squamous cell carcinoma ((HNSCC) samples ($n = 22$))[43,46]. We chose HPV-negative HNSCC as controls because they have a sufficiently high rate of indels and structural variants (SVs) to provide the required statistical power.

For these two cohorts, we called SVs with BRASS[43] and annotated by AnnotateBRASS (https://github.com/MathijsSanders/AnnotateBRASS). We determined the following statistics per SV: the number of supporting read-pairs, the alignment position variance of supporting read-pairs, the frequency of read clipping, the frequency of reads with an excess of variants (≥2) absent from dbSNP, the proportion of read-pairs correctly oriented based on the SV detection and the number of SV-supporting read-pairs proximal to the SV breakpoints with alternative alignments (high genome homology). The post-annotation filtering strategy was previously described in detail (https://github.com/cancerit/BRASS). We analyzed the PCAWG-HNSCC (BRCA2-proficient) and PCAWG-BRCA2$^{mut}$ (BRCA2-deficient) utilizing the same methodology.

Additionally, we counted the total number of long microhomology-assisted deletions (size range 1.4 kb–272.9 kb, 95% interval) contained within either a cLAD or ciLAD. We tested if the number of MH deletions were differently distributed between cLADs

and ciLADs between cohorts by using bootstrapping of the controls, as described previously for short deletions (Supplementary Data 7).

### Reporting summary

Further information on research design is available in the Nature Portfolio Reporting Summary linked to this article.

## Data availability

All raw data shown in this paper have been deposited in the NCBI's Sequence Read Archive (SRA) and are accessible through the BioProject number PRJNA882344. Processed data is available as Supplementary Data and/or can be found on GitHub (https://github.com/vansteensellab/CCD_repair_protein_project/tree/main/data/processed_data)[61]. GEO accession numbers and links to chromatin profiling data used in this manuscript can be found in Supplementary Data 3. Sporadic human tumor data is available from TCGA (https://portal.gdc.cancer.gov/projects), PCAWG and the full list of driver mutations identified by the pan-cancer analysis of whole genomes (PCAWG) consortium was obtained for all available cancer subtypes (https://dcc.icgc.org/releases/PCAWG/driver_mutations). The datasets and tumors identifiers used in for this manuscript are listed in Supplementary Data 7. Source data are provided with this paper.

## Code availability

All code to analyze the data and create the figures is available on GitHub (https://github.com/vansteensellab/CCD_repair_protein_project)[61].

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

## Acknowledgements

We thank the NKI Genomics and Research High Performance Computing core facilities for technical support; members of our laboratories and Titia Sixma for inspiring discussions and helpful comments; Jacqueline

Jacobs, Heinz Jacobs and Jeroen van den Berg for help with KO library gene set curation. This work was supported by ZonMW TOP grant 91215067 (to R.H.M. and B.v.S.), European Research Council (ERC) Advanced Grant 694466 (to B.v.S); NIH Common Fund "4D Nucleome" Program grant U54DK107965 (B.v.S.); NWO Zwaartekracht (to R.H.M.); KWF infrastructure grant 12539 (to R.L.B.). The Oncode Institute is partly supported by KWF Dutch Cancer Society.

## Author contributions

X.V., A.G.M., R.S., B.v.S., R.H.M. conceived and designed the study. A.G.M., M.d.H., B.M., A.F. and X.V. performed experiments. X.V., C.L. and M.S. performed data analysis. B.v.S., R.H.M. and R.L.B. supervised the study. X.V., B.v.S. and R.H.M. wrote the manuscript with input of all authors.

## Competing interests

The authors declare no competing interests.
