## [Peer Review File · Nature Communications]

Widespread chromatin context-dependencies of DNA double-strand break repair proteinsEditorial Note: This manuscript has been previously reviewed at another journal that is not operating a transparent peer review scheme. This document only contains reviewer comments and rebuttal letters for versions considered at *Nature Communications* .

REVIEWERS' COMMENTS

Reviewer #1 (Remarks to the Author):

I thank the authors for their detailed response and helpful clarifications. I agree that the unusually low editing efficiency mainly resulted in a very low dynamic range in the primary screen, while the signal-to-noise ratio seems acceptable based on the FDR estimates. Nevertheless, I hope the authors agree that higher CRISPR editing efficiencies would have been desirable, and I hope they will consider incorporating more efficient editing strategies prior to future screens. Regarding validation experiments, I agree that the use of RNAi or small molecule inhibitors as orthogonal approaches adds some value, but I disagree with their notion that "full knockouts are challenging in K562 cells because they are triploid/tetraploid". Both published CRISPR-based knockout screens and our own experience clearly show that state-of-the-art CRISPR systems do indeed allow very efficient KO generation in K562 cells in bulk, and I strongly encourage the authors to consider this for future studies. Although it is somewhat disappointing that such clean CRISPR-based validations for K562 cells are still missing, the RNAi and small molecule-based validations provided are sufficient. Whether or not to include the problematic data in DDR-proficient RPE-1 cells should be at the discretion of the authors.

Overall, the manuscript by Vergara et al. uses an innovative screening approach to provide comprehensive 'systems-level' insight into the interesting and so far unexplored question of how chromatin context affects DNA repair mechanisms. The manuscript incorporates a large body of work and screening data that provides a valuable starting point for further exploration of this question, both at the systems level and for individual factors in greater depth.

Response to reviewer

Reviewer #1 (Remarks to the Author):

I thank the authors for their detailed response and helpful clarifications. I agree that the unusually low editing efficiency mainly resulted in a very low dynamic range in the primary screen, while the signal-to-noise ratio seems acceptable based on the FDR estimates. Nevertheless, I hope the authors agree that higher CRISPR editing efficiencies would have been desirable, and I hope they will consider incorporating more efficient editing strategies prior to future screens. Regarding validation experiments, I agree that the use of RNAi or small molecule inhibitors as orthogonal approaches adds some value, but I disagree with their notion that "full knockouts are challenging in K562 cells because they are triploid/tetraploid". Both published CRISPR-based knockout screens and our own experience clearly show that state-of-the-art CRISPR systems do indeed allow very efficient KO generation in K562 cells in bulk, and I strongly encourage the authors to consider this for future studies. Although it is somewhat disappointing that such clean CRISPR-based validations for K562 cells are still missing, the RNAi and small molecule-based validations provided are sufficient. Whether or not to include the problematic data in DDR-proficient RPE-1 cells should be at the discretion of the authors.

Overall, the manuscript by Vergara et al. uses an innovative screening approach to provide comprehensive 'systems-level' insight into the interesting and so far unexplored question of how chromatin context affects DNA repair mechanisms. The manuscript incorporates a large body of work and screening data that provides a valuable starting point for further exploration of this question, both at the systems level and for individual factors in greater depth.

We thank the reviewer for the feedback and we are glad that our response helped. For the reasons explained in previous rebuttal, we will not include the DDR-proficient RPE-1 cell data.

Brief summary

DNA double-strand breaks are repaired by multiple pathways. The balance of these pathways depends on the local chromatin context, but the underlying mechanisms are poorly understood. Here the authors uncover a network of proteins that regulate pathway balance in a chromatin context-dependent manner.